# Just-Noticeable Differences of Fundamental Frequency Change in Mandarin-Speaking Children with Cochlear Implants

**DOI:** 10.3390/brainsci12040443

**Published:** 2022-03-26

**Authors:** Wanting Huang, Lena L. N. Wong, Fei Chen

**Affiliations:** 1Department of Electrical and Electronic Engineering, Southern University of Science and Technology, Shenzhen 518055, China; huangwt@sustech.edu.cn; 2Unit of Human Communication, Development, and Information Sciences, Faculty of Education, The University of Hong Kong, Hong Kong 999077, China; llnwong@hku.hk

**Keywords:** cochlear implants, children, fundamental frequency, demographic factor

## Abstract

Fundamental frequency (F0) provides the primary acoustic cue for lexical tone perception in tonal languages but remains poorly represented in cochlear implant (CI) systems. Currently, there is still a lack of understanding of sensitivity to F0 change in CI users who speak tonal languages. In the present study, just-noticeable differences (JNDs) of F0 contour and F0 level changes in Mandarin-speaking children with CIs were measured and compared with those in their age-matched normal-hearing (NH) peers. Results showed that children with CIs demonstrated significantly larger JND of F0 contour (JND-C) change and F0 level (JND-L) change compared to NH children. Further within-group comparison revealed that the JND-C change was significantly smaller than the JND-L change among children with CIs, whereas the opposite pattern was observed among NH children. No significant correlations were seen between JND-C change/JND-L change and age at implantation /duration of CI use. The contrast between children with CIs and NH children in sensitivity to F0 contour and F0 level change suggests different mechanisms of F0 processing in these two groups as a result of different hearing experiences.

## 1. Introduction

Fine pitch processing is relatively scarce in non-tonal languages, whereas in tonal languages, which account for 70% of the world’s languages [1], rapid pitch variations (i.e., lexical tones) are used to alter the meaning of a syllable. For example, in Mandarin Chinese, each of the four lexical tones has a distinct pattern of pitch inflection: level, mid-rising, dipping, and high-falling, changing the meaning of a syllable such as /ma/ into “mother”, “hemp”, “horse”, or “scold”. Prior research findings have consistently suggested that fundamental frequency (F0) provides the primary acoustic information for Mandarin tone recognition [2], while the temporal [3,4] and spectral envelope [5] serve as the secondary acoustic cues.

Cochlear implantation is widely accepted as a life-changing invention for individuals with severe to profound hearing impairment. According to the Sixth National Population Census of the People’s Republic of China, the total number of hearing-impaired children aged 0 to 14 was estimated to be more than 4.6 million [6]. It is reported that about 2000 preschool children with congenital severe to profound hearing loss in mainland China (hereafter referred to as Mandarin-speaking children with CIs) underwent cochlear implantation in 2004, increasing at a rate of 30% to 50% per year thereafter [7].

The speech processing strategies currently used in CIs are mainly designed to encode the temporal envelope of sound stimuli. Given the vulnerability of the temporal envelope in noise, it is not surprising that speech perception in noise conditions among Mandarin-speaking children with CIs remains unsatisfactory [8,9] and is even worse than that in English-speaking children with CIs [10,11]. In recent decades, to improve speech perception in CI users speaking tonal languages, speech processing strategies have been developed to provide more F0 information by delivering more spectral information (e.g., HiRes 120) or by enhancing temporal fine structure cues (e.g., Temporal Fine Structure) in CI systems. Unfortunately, these new speech processing strategies have not improved tone perception in CI users who speak tonal languages [12,13]. For instance, using a two-alternative forced-choice paradigm, researchers found that two speech processing strategies (i.e., HiRes or HiRes 120) failed to produce statistically significant differences in Mandarin tone recognition performance among a group of 20 Mandarin-speaking children with CIs, although most of them reported a preference for HiRes 120 [12]. Such negative results have also been found among Cantonese-speaking adult CI users when comparing Cantonese tone recognition using a typical speech processing strategy (i.e., continuous interleaved sampling) to that using a relatively new one [13].

The insignificant improvement in lexical tone recognition using newer speech processing strategies, as reported in previous studies, might have been the result of a lack of understanding of F0 processing in Mandarin-speaking CI users, which has received little attention until recent years. F0s of Mandarin tones comprise two dimensions: F0 level and F0 contour. F0 level, which refers to the height of onset F0 in Mandarin tones, is usually used to identify different talkers (e.g., male vs. female), while F0 contour reflects the trajectory of F0 change over the duration of a single tone and plays a more decisive role in the discrimination of word meaning compared to F0 level. Existing studies on F0 contour processing in Mandarin-speaking children with CIs suggest that these children may use F0 contours differently from their age-matched NH peers as a result of different hearing experiences [14,15]. For example, researchers found that, although Mandarin-speaking children with CIs were able to use F0 contours for tone recognition, they tended to rely more on the temporal envelope than on F0 contours when performing word-level tone recognition tasks [15]. However, the situation was quite different at the sentence level. A recent study investigated the effects of F0 contours on sentence recognition in Mandarin-speaking preschool children with CIs in both quiet and noise conditions. The results showed that when other acoustic cues (e.g., temporal envelope) were neutralized, sentence recognition with flattened F0 contours was significantly worse than that with normal F0 contours in both children with CIs and NH children. While the F0 contour-caused decrease in sentence recognition was only seen in quiet conditions among the NH children, it was seen in both quiet and noise conditions among the children with CIs. Furthermore, the impact of F0 contours on sentence recognition accuracy in children with CIs was significantly more salient than that in NH children [14].

Processing of the F0 level in Mandarin-speaking children with CIs has received much less attention compared to that of the F0 contour. Similar to previous studies on music perception in CI users [16,17], the few studies on F0 level processing in Mandarin-speaking children with CIs have demonstrated deficits when compared to NH controls. For example, researchers measured and compared F0 level discrimination in 24 Mandarin-speaking school-aged children (aged 4.6 to 21.3 years) and found that percent correct performance for F0 level discrimination in the CI group was significantly poorer than that in the NH group, suggesting a compromise in F0 level processing in Mandarin-speaking school-aged children with CIs [18].

The extent to which sensitivity to F0 contour and F0 level change is affected by CI-related demographic factors (e.g., age at implantation, duration of CI use) indicates whether and how F0 processing in Mandarin-speaking children with CIs is shaped by the children’s experiences on using CIs. Currently, the relationship between sensitivity to F0 change and CI-related demographic factors is rarely reported in the existing literature. According to the existing literature, while reliance on temporal envelope was significantly correlated with age at implantation, there was no significant relationship between reliance on F0 contours and the duration of CI use [15]. In English-speaking preschool children, researchers found no significant correlations between CI-related demographic factors and sensitivity to F0 contour change [19]. Similarly, no significant correlations between CI-related demographic factors (i.e., age at implantation and duration of CI use) and sensitivity to F0 level change were reported in either English-speaking or Mandarin-speaking preschool children [18].

To date, there is still a lack of understanding of F0 contour and F0 level processing in Mandarin-speaking children with CIs and of the extent to which F0 processing in these children is similar to or different from that in age-matched NH children. In the present study, just-noticeable differences (JNDs) of F0 contour and F0 level change, which were used as indicators of F0 processing, were measured and compared between Mandarin-speaking kindergarten-aged children with normal hearing and children with CIs. Given the well-known deficits in F0 processing of speech sounds in CI systems, it was predicted that children with CIs would be significantly less sensitive to both F0 level and F0 contour change compared to NH children. Based on the previously reported larger JND of pitch contour change in NH adults compared to the JND of pitch level change [20], it was predicted that in NH children, the JND of F0 contour change would also be larger than that of F0 level change. As F0 contour and F0 level perception have only been examined separately in previous studies, the relative sensitivity to F0 contour change and F0 level change has yet to be explored.

To gain a better understanding of the relationship between CI-related demographic factors (e.g., age at implantation, duration of CI use) and F0 change detection in Mandarin-speaking children with CIs, correlation analyses were conducted between CI-related demographic factors and JNDs of F0 contour and F0 level change in the Mandarin-speaking children with CIs. Based on the previously reported results on this issue, it was expected that CI-related demographic factors would not be significantly correlated with the JNDs of F0 contour and F0 level change in the Mandarin-speaking kindergarten-aged children with CIs.

## 2. Materials and Methods

### 2.1. Participants

Thirty Mandarin-speaking preschool children with CIs (18 males and 12 females; mean age 4.36 ± 0.70 years old) participated in the current study. The preschool children were chosen for the following reasons: (a) children with normal hearing exhibit protracted development of tonal processing before school age [21,22,23,24]; (b) there is a critical period for central auditory system in pediatric CI users before the age of 7 [25,26]; (c) most prelingually deafened children in mainland China receive implantation before the age of six. Therefore, investigations in children before school age provide evidence of tonal development at an early stage, which offers valuable references for the development and improvement of the tonal language-oriented speech processing strategies in CI systems. All participants were recruited from the Beijing Children’s Hospital and the China Rehabilitation Research Center for Hearing and Speech Impairment. Children in the CI group met the following inclusion criteria: (a) aged 3–6 years, (b) diagnosed with congenital bilateral severe to profound sensorineural hearing impairment, (c) had received unilateral implantation, and (d) had been using CIs for not less than six months. Children in this group were using CIs from four manufacturers: Advanced Bionics (*n* = 5), Cochlear (*n* = 12), MED-EL (*n* = 12), and Nurotron (*n* = 1). The speech processing strategies used by these CI manufacturers are HiResolution (HiRes) 120, Advanced Combination Encoder (ACE), Fine Structure (FS) 4, and C-tone, respectively. Among these children, 12 had undergone a unilateral (*n* = 5) or bilateral (*n* = 7) hearing aid trial before implantation, and 25 wore a hearing aid on the non-implanted ear after implantation. Details of demographic information and device use are shown in Table 1. Thirty age-matched Mandarin-speaking children (17 male and 13 female, mean age: 4.37 ± 0.48 years old) with audiometric thresholds not worse than 20 dB HL at octave frequencies between 250 and 4000 Hz were included as normal controls in this study. Children in both groups scored within the normal range on the Hiskey-Nebraska Test of Learning Aptitude for children above 3 years of age [27]. The research study was approved by the Human Research Ethics Committee of the University of Hong Kong and Beijing Children’s Hospital. All children participated voluntarily in the study, with informed consent obtained from their parents.

### 2.2. Stimuli

The original stimulus, an isolated Mandarin vowel /a/ with tone 1, was first spoken by an adult female native Mandarin speaker. Using Praat [28], the F0 contour of /a1/ was then replaced with a series of linear F0 contours, with other acoustic features remaining the same. JNDs of F0 contour and F0 level change were measured in two separate blocks. In the block measuring the JND of F0 contour change, the offset F0s were manipulated to vary from 100 to 300 Hz, with the onset of the linear F0 contours being fixed at 100 Hz, resulting in an offset continuum with F0 contours ranging from a level tone to a rising tone. The step size between adjacent offset F0s was set at 1 Hz, and thus, the newly resynthesized stimuli were made up of 201 /a/ carrying different F0 contours (see Figure 1A). The stimuli in the block measuring the JND of F0 level change were the same as those in the F0 contour condition, except that the onset F0 was equal to the offset F0 throughout the resynthesized F0 contours (see Figure 1B). According to the Syllabus of the Chinese Proficiency Test, the duration of a naturally uttered lexical tone is around 200 to 300 ms [29]. To make the stimuli more natural and to guarantee the audibility of stimuli, the duration of each stimulus in both conditions was set at 300 ms.

### 2.3. Procedure

A three-alternative forced-choice paradigm with a two-down, one-up tracking algorithm was used to measure the JNDs of F0 contour and F0 level change among the children in both groups. Within each trial, two standard stimuli and one deviant stimulus were randomly presented, with the inter-stimulus interval being 400 ms. The probability of the deviant stimulus appearing at each interval was equal to 1/3. In the F0 level condition, 100–100 Hz was selected as the standard stimulus. In contrast, to ensure that pitch contour was the main cue for the detection of F0 change and not pitch height, three flat contours (i.e., 100–100, 200–200, and 300–300 Hz) were used as standard stimuli in the F0 contour condition, two of which were randomly chosen in each trial.

During the test, children in the CI group wore their CIs only. Children in both groups were seated at a rectangular table in a quiet room while performing the task. The sound stimuli were presented at a listening level of 65 dB SPL via a loudspeaker located in front of the children at a 0° azimuth and a distance of one meter from the center of the head of participants. Three identical cartoon dogs were printed on three separate pieces of paper. During the presentation of the three sound stimuli in each trial, the examiner was seated next to the children and pointed at the cartoon dogs one by one with the sound stimuli. Children were asked to indicate which dog’s voice sounded different from the other two (see Figure 2). In the F0-contour block, children were first taught with the targeted rising tone (i.e., 100–300 Hz). In each trial of this block, children were presented with the rising tone three times before the onset of sound stimuli.

Each of the two blocks contained 60 trials. In both blocks, the offset F0s of the deviant stimuli were first started at 300 Hz and went downward and approached the standard stimulus (100 Hz) upon correct responses. Following the successful discrimination of the deviant stimulus in the first trial, the offset F0 change was set at 100 Hz for the second trial. Based on previous studies [20,30], the step size in each trial was adjusted to 5 Hz for the first three reversals and to 1 Hz thereafter. With the exclusion of the first three reversals, the average offset F0 change from the original one (300 Hz) was calculated from the last even number of reversals in the adaptive track. The JND for each condition was defined as the offset F0 difference between the average offset F0 change and the offset F0 of the standard stimulus (i.e., 100 Hz). The children were given practice trials before the test until they were familiarized with the task requirements. The order of the blocks of F0 contour and F0 level measurement was counterbalanced among the children. A break was given every four to six trials.

### 2.4. Statistical Analysis

To investigate the effects of hearing experience (normal hearing vs. CIs) and F0 dimensions on the sensitivity to F0 change, JNDs of F0 contour change and F0 level change were entered as the dependent variables in a two-way analysis of variance, with group (NH group/CI group) and F0 dimension (F0 level/F0 contour) as the independent variables. A test of simple effects was conducted upon the significant interaction between group and F0 dimension. The Mann–Whitney U test was performed for the comparison between groups.

To investigate the relationship between demographic factors (e.g., age at implantation) and sensitivity to F0 change, Pearson correlation analyses were performed between JNDs of F0 contour/ F0 level change and demographic factors.

The *p*-value of 0.05 was set as a threshold of statistical significance throughout all tests.

## 3. Results

The JNDs of F0 contour and F0 level change in children with CIs and NH children are shown in Figure 3. On the whole, the JNDs of F0 change were larger in children with CIs than in their age-matched peers. Within-group comparison between the JND of F0 level change and the JND of F0 contour change revealed that, in the control group, the JNDs of F0 contour change were consistently larger than those of F0 level change, while in children with CIs, contrasting patterns were observed: 22 of the 30 children with CIs exhibited larger JNDs of F0 level change, 6 showed opposite patterns, and 2 demonstrated equal JNDs of F0 contour and F0 level change. The percentage of children exhibiting different patterns of JNDs of F0 level change and F0 contour change for each type of speech processing strategy is shown in Table 2.

Significant positive correlations between the JND of F0 contour change and the JND of F0 level change were found in the NH group (r = 0.622, *p* < 0.001), the CI group (r = 0.636, *p* < 0.001), and the combination of the two groups (r = 0.793, *p* < 0.001). A two-way analysis of variance (ANOVA), with group (children with CIs/normal hearing) as the between-subject factor and F0 dimension (F0 level/contour) as the within-subject factor (see Figure 4), demonstrated the significant main effects of group (F (1, 58) = 102.12, *p* < 0.001, partial η^2^ = 0.64) and F0 dimension (F (1, 58) = 21.17, *p* < 0.001, partial η^2^ = 0.27) and interaction between group and F0 dimension (F (1, 58) = 4.54, *p* < 0.001, partial η^2^ = 0.57). The ANOVA results are summarized in Table 3. Post hoc analysis revealed significantly larger JNDs of F0 contour change (F (1, 58) = 38.07, *p* < 0.001, partial η^2^ = 0.40) and F0 level change (F (1, 58) = 141.43, *p* < 0.001, partial η^2^ = 0.0.71) in children with CIs than those in NH children. Among NH children, the JND of F0 contour change was significantly larger than that of F0 level change (F (1, 58) = 92.50, *p* < 0.001, partial η^2^ = 0.61). In contrast, among children with CIs, the JND of F0 contour change was found to be significantly smaller than the JND of F0 level change (F (1, 58) = 8.67, *p* < 0.01, partial η^2^ = 0.13). Furthermore, an independent-samples *t*-test, comparing the difference between the JND of F0 contour change and the JND of F0 level change in both groups, suggested that the JND difference between F0 contours and F0 levels was significantly larger in NH children than in children with CIs (t (58) = 4.72, *p* < 0.001).

No significant correlation was observed between age at implantation and either the JND of F0 contour change (r = −0.09, *p* = 0.65) or the JND of F0 level change (r = −0.15, *p* = 0.42) or between duration of CI use and either the JND of F0 contour change (r = −0.09, *p* = 0.64) or the JND of F0 level change (r = 0.03, *p* = 0.86).

## 4. Discussion

The main purpose of this study is to investigate the processing of F0, including F0 level and F0 contour, in Mandarin-speaking children with CIs as compared to NH peers, which may partially account for the unsatisfactory outcome of Mandarin tone recognition in CI users. To achieve this goal, we investigated the JNDs of F0 contour change and F0 level change in Mandarin-speaking kindergarten-aged children with CIs compared to those of their age-matched NH peers. The results showed that NH children were more sensitive to F0 level changes, as evidenced by the significantly smaller JND in the F0 level condition than that in the F0 contour condition. The higher sensitivity to F0 level change compared to sensitivity to F0 contour change in NH children is consistent with previous findings among NH adults at both behavioral [20] and electrophysiological levels [31]. It should be noted that, although the F0 onset and offset and measurement methods in the F0 level condition in the present study (100 Hz to 300 Hz) were similar to those in NH adults (180 to 250 Hz) [20], the JNDs of F0 level change obtained among NH children in this study was much larger than those that have been found among NH adults [19,20]. The relatively larger JNDs in NH children compared to NH adults are probably the result of the children’s less-developed central auditory system, which does not fully mature until about 12 years of age [32,33]. However, since the measurement of JND of F0 level change and JND of F0 contour change was not fully equivalent, it may not be suitable to conclude the relationship between the sensitivity to F0 level change and that to F0 contour change in this group alone.

Although positive correlations between the JNDs of F0 contour and F0 level change were found in both groups of children in the present study, the sensitivity to F0 change in children with CIs was quite different from that in NH children. While NH children consistently showed higher sensitivity to F0 level changes compared to F0 contour changes, large individual variabilities were observed in the CI group. Although the majority of children in the CI group (*n* = 22) exhibited larger JNDs for F0 level change, four children exhibited the opposite pattern, while two children showed equal sensitivity to F0 contour and F0 level change. Substantial individual variability in speech perception in Mandarin-speaking CI users has been reported in many studies [12,34], which probably results from variations in demographic factors, such as age at implantation [12] and duration of CI use [35], although the results of the present study did not show significant correlations between the JNDs of F0 contour and F0 level change and these demographic factors. This issue is discussed later.

It is known that in CIs, F0 information up to approximately 300 Hz is processed by way of temporal information, which conveys far less fine structure information than can be processed by the normal auditory system [36,37]. Therefore, it is not surprising that children with CIs show a deficit in F0 processing, as reflected by the significantly larger JNDs of both F0 contour and F0 level change compared to NH children. These findings are consistent with those reported in previous studies [15,18,19]. More importantly, as shown in Figure 4, the relationship between sensitivity to F0 contour change and to F0 level change in children with CIs was in contrast to that in the NH children at the group level. While the NH controls demonstrated higher sensitivity to F0 level change, the pediatric CI users were more sensitive to F0 contour change. It is worth noting that the higher sensitivity to F0 contour change compared to F0 level change was widely seen in the CI group, regardless of the type of speech processing strategies used by these children (see Table 2). Therefore, the significant sensitivity to F0 contour change observed in the CI group cannot simply be attributed to speech processing strategies; rather, it is probably a common phenomenon in CI users, resulting from their hearing experience with CIs. Such higher sensitivity to F0 contour change compared to F0 level change could be caused by two factors. First, as reported by the previous study, amplitude modulation depth is usually inconsistently coded by clinical speech processing strategies (even for those produced by the same talker), which results in the inconsistent perception of the F0 level; however, the perception of the F0 contour is not likely to be affected [38]. Under such circumstances, it is not surprising that the inconsistent perception of the F0 level leads to a significantly larger JND of F0 level change than that of F0 contour change. Second, as introduced above, CIs are designed to extract temporal envelope information so that the contour of the frequency fluctuations of speech is maintained. To master a tonal language, in which contour information plays a dominant role in discriminating word meanings, CI users must maximally utilize the contour information conveyed by CIs. Thus, children with CI users in the present study might have developed a unique mechanism of F0 processing that differs from that of their NH peers. In other words, the contour information of speech sounds might have been prioritized compared to F0 level information in speech perception among children with CIs. The difference in sensitivity to F0 level and F0 contour change between children with CIs and NH children suggests that the enhancement of F0 information in the newly developed speech processing strategies (e.g., HiRes 120, Temporal Fine Structure) in Mandarin-speaking children with CIs may not fully satisfy the requirements for F0 processing in speech perception, which partially explains the unsatisfactory improvement in speech perception among this population when switching to new speech processing strategies [12].

No significant correlation was found between JNDs of F0 contour/F0 level and age at implantation/duration of CI use in this study. This finding was consistent with the previous study, in which 23 English-speaking school-aged children with CIs were evaluated [19]. Children in both studies were prelingually deaf, and their hearing during tests depended on the implanted CI on either side of the ear, with the hearing aid on the contralateral side being turned off and removed from the ear. The insignificant relationship between JNDs of F0 contour/F0 level and age at implantation/duration of CI use in this study suggests that age at implantation and duration of CI use may have little impact on the F0 processing in these children. However, it is also possible that given the relatively small sample size, the effect of age at implantation and duration of CI use was overwhelmed by the well-recognized individual variability [39].

There are several limitations in the current study. According to a recent study, different types (i.e., the posterior tympanotomy technique vs. the endomeatal approach) of CI surgery result in different levels of postoperative discomfort [40]. Unfortunately, the type of surgery in the CI group was not well documented in the present study, and it was unclear whether the type of surgery would play a role in the F0 change detection in children with CIs. Technically speaking, the round window approach manages to preserve hearing residues [40] so that F0 change detection in CI recipients will be better. Such an inference is expected to be verified by further studies. In addition, the relatively small sample size (*n* = 30) in this study made it unlikely to demonstrate the relationship between demographic factors and F0 processing in children with CIs. A larger sample size is required to address this question in the future. Additionally, given that the F0 level and F0 contour were processed holistically when perceiving tones [41], an improved paradigm is needed to further confirm the findings in this study.

## 5. Conclusions

The current study explored F0 processing in Mandarin-speaking children with CIs from a psychophysical perspective. The results revealed significantly compromised sensitivity to F0 change in children with CIs. Specifically, children with CIs demonstrated higher sensitivity to F0 contour change than to F0 level change; this pattern contrasted with the relatively well-developed sensitivity to F0 level change in NH children. However, CI-related demographic factors (i.e., age at implantation and duration of CI use) seemed to have little impact on sensitivity to F0 change in children with CIs. This is the first study to investigate F0 processing in Mandarin-speaking preschool children with CIs by comparing the two dimensions of F0 (i.e., F0 contour and F0 level). The contrast in sensitivity to F0 contour and F0 level change between children with CIs and NH children may suggest different mechanisms of F0 processing in these two groups as a result of their hearing experiences. To the best of our knowledge, this is the first study exploring both dimensions of F0 information in Mandarin-speaking children with CIs. The difference in F0 processing between children with CIs and NH children, revealed in this study, provides new perspectives on the development and improvement of speech processing strategies in CI systems, especially those targeted at tonal language speakers. It is believed that it will significantly improve the life quality of CI users by providing more accurate F0 information in the CI systems.

## Figures and Tables

**Figure 1 brainsci-12-00443-f001:**
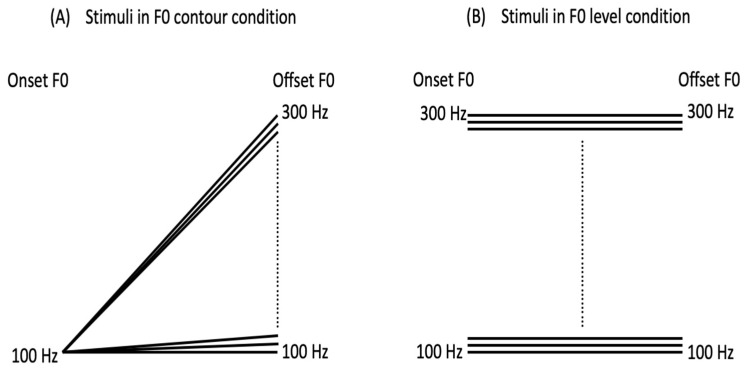
Schematic representation of linear F0 contours in F0 contour (**A**) and F0 level (**B**) conditions.

**Figure 2 brainsci-12-00443-f002:**
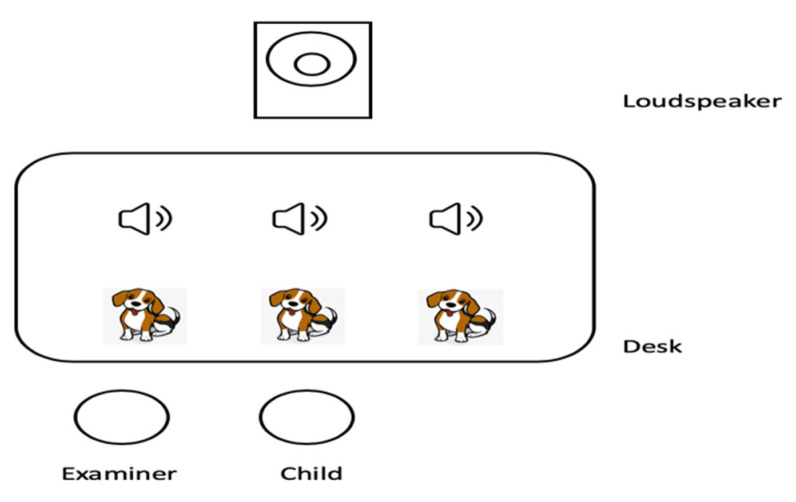
Diagram of the experimental scene. The loudspeaker is one meter away from the child.

**Figure 3 brainsci-12-00443-f003:**
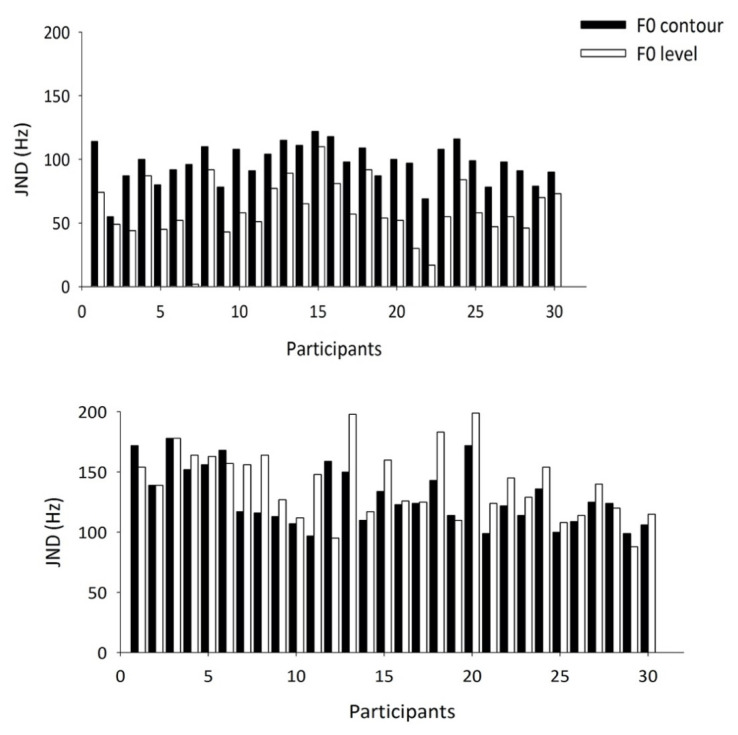
JNDs of F0 contour change and F0 level change in the 30 children with normal hearing (**upper panel**) and in the 30 children with CIs (**lower panel**).

**Figure 4 brainsci-12-00443-f004:**
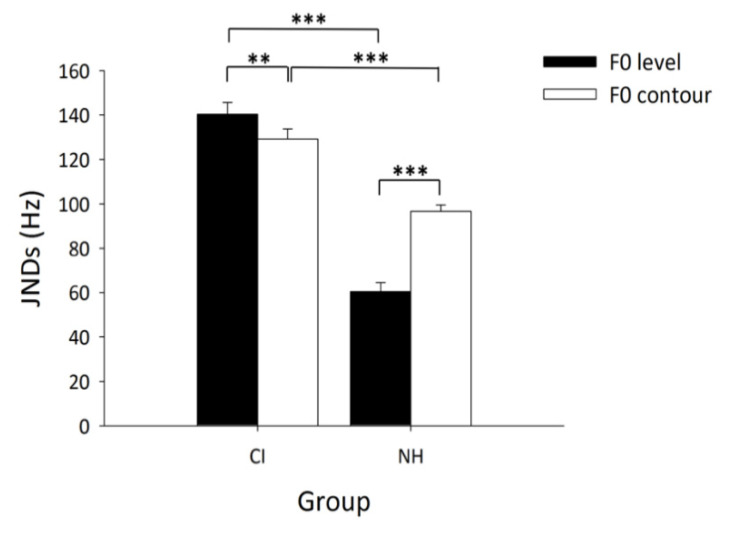
JNDs as a function of F0 dimensions in the CI and NH groups. Error bars indicate standard errors. The level of significance is indicated with asterisks (**: *p* < 0.01; ***: *p* < 0.001).

**Table 1 brainsci-12-00443-t001:** Demographics of children with CIs.

No.	Sex	AAT (Years)	HAT (Years)	AAI (Years)	Hearing Device			DCI (Years)	DAVT (Years)
					Left Device	Right Device	CI Speech Processing Strategies		
1	F	5.75	1.75	3.58	Phonak	MED-EL	FS4	2.17	2.17
2	F	5.08	0.42	3.33	Phonak	Cochlear	ACE	1.75	1.75
3	M	4.33	0.33	1.42	Phonak	AB	HiRes120	2.91	2.91
4	F	4.83	0.83	2.92	Cochlear	Phonak	ACE	1.91	1.75
5	M	4.92	3.67	3.67	Cochlear	Widex	ACE	1.25	1.25
6	M	5.33	0	3.17	Phonak	MED-EL	FS4	2.16	2.16
7	M	3.92	0	1.33	Phonak	MED-EL	FS4	2.59	2.5
8	F	4.83	0	2.58	Phonak	Nurotron	C-tone	2.25	1.91
9	F	4.08	1	3.33	MED-EL	Phonak	FS4	0.75	0.75
10	F	4.08	0	2.58	MED-EL	Phonak	FS4	1.5	1.33
11	F	4	0.33	2.75	Phonak	Cochlear	ACE	1.25	1.25
12	M	4.17	0	3	MED-EL	Phonak	FS4	1.17	1.17
13	M	4.33	0.75	3.25	Phonak	Cochlear	ACE	1.08	1.08
14	M	3.75	0	2	AB	Phonak	HiRes120	1.75	1.5
15	M	4.25	0	2.25	Phonak	Cochlear	FS4	2	2
16	M	3.5	0	1.67	Phonak	MED-EL	FS4	1.83	1.25
17	F	4.17	0.67	1.58	Cochlear	Widex	ACE	2.59	1.59
18	M	3.83	0.67	1.92	Cochlear	Phonak	ACE	1.91	1.91
19	M	4	0	2.08	Phonak	Cochlear	ACE	1.92	1.67
20	M	3.58	0	0.67	Phonak	AB	HiRes120	2.91	2.08
21	M	3.5	0	1.58	Phonak	Cochlear	ACE	1.92	1.92
22	F	3.42	0.67	2.08	MED-EL	Phonak	FS4	1.34	1.34
23	M	3.5	0	1.83	Cochlear	Phonak	ACE	1.67	1.5
24	M	4.08	0	2.08	Cochlear	Phonak	ACE	2	1.5
25	F	5.75	0.92	2.5	AB	Null	HiRes120	3.25	1.75
26	M	4.92	0	1.67	MED-EL	Null	FS4	3.25	1.33
27	F	4.33	0	1.17	Null	MED-EL	FS4	3.17	1.5
28	F	5.5	0	2.17	AB	Null	HiRes120	3.33	0.92
29	M	5.33	0	4.17	Phonak	MED-EL	FS4	1.17	0.83
30	M	3.83	0	2.08	Cochlear	Null	ACE	1.53	1.67

AAT: age at test; HAT: hearing aid trial before implantation; AAI: age at implantation; DCI: duration. of CI use; DAVT: duration of auditory-verbal training; AB: Advanced Bionics; FS4: Fine Structure 4; ACE: Advanced Combination Encoder; HiRes 120: HiResolution 120.

**Table 2 brainsci-12-00443-t002:** Percentage of the children with CIs exhibiting different patterns of JNDs of F0 level (JND-L) and F0 contour (JND-C) change regarding each type of speech processing strategy.

CI Speech Processing Strategies	JND-L > JND-C	JND-L = JND-C	JND-L < JND-C
FS4 (*n* = 12)	66.7%	0%	33.3%
ACE (*n* = 12)	83.3%	8.3%	8.3%
HiRes 120 (*n* = 5)	60%	20%	20%
C-tone (*n* = 1)	100%	0%	0%

CI: cochlear implant; FS4: Fine Structure 4; ACE: Advanced Combination Encoder; HiRes 120: HiResolution 120.

**Table 3 brainsci-12-00443-t003:** Summary of the ANOVA results.

Source of Variation	df	F Value	Partial Eta^2^
group	1	102.12 ***	0.64
F0 dimension	1	21.17 ***	0.27
group × F0 dimention	1	4.54 ***	0.57

Eta^2^: Eta squared; ***: *p* < 0.001.

## Data Availability

Data are available upon request from the corresponding authors.

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
