# Peer review of "Just-Noticeable Differences of Fundamental Frequency Change in Mandarin-Speaking Children with Cochlear Implants"

_brainsci, 2022, doi:10.3390/brainsci12040443_

Round 1

Reviewer 1 Report

The authors wrote an article about the differences of fundamental frequency in Mandarin-speaking children with cochlear implants. The main theme should be interesting, but key parts of the article structure are missing. Here are the corrections I suggest you make.
Methods
1. The type of surgery performed is missing. Has the array been accessed in a round window? A promontory cochleostomy? Was the classic posterior tympanotomy or endomeatal access performed?
There are some differences in the result between a traditional surgery and a soft surgery, consult the following reference: Freni F, Gazia F, Slavutsky V, Scherdel EP, Nicenboim L, Posada R, Portelli D, Galletti B, Galletti F. Cochlear Implant Surgery: Endomeatal Approach versus Posterior Tympanotomy. Int J Environ Res Public Health. 2020 Jun 12;17(12):4187. 
2. The paragraph "statistical analysis" is missing, where all the statistics performed, and the tests used are summarized. I noticed that they were described in the results correctly, but it takes the paragraph in the methods, as per the editorial guidelines.
Discussion / Conclusion
1. The limitations section of the study is missing. For example, do demographic factors influence F0 ?, Is the sample not limited?
2. What is the main purpose of the study? Did the change in F0 in implanted children, affect them in the quality of life? in the psychological aspect? Does the child neuropsychiatrist have to be involved?
3. What novelty does study bring to literature? Emphasize the conclusions 

Reviewer 2 Report

The manuscript describes a study of the perception of changes in f0 in Mandarin-speaking children with cochlear implants (CI).  Encoding and perception of f0 is especially important for speakers of tone languages, of which there are many.  Two kinds of f0 or pitch change were studies:  a simple change in steady-state f0 (called f0 level) and a change in the end point of an f0 glide (called f0 contour).  CI children had slightly-lower thresholds for f0 contour change.  Normal-hearing children had slightly-lower thresholds for f0 level, the opposite pattern.  Threshold of CI children were significantly higher than threshold of NH children for both conditions.  The methods are appropriate, and the data are clearly presented. The results may have implications for strategies for CI processing for speakers of tone languages, so this is an important area of research.  

Author Response

Dear Reviewer, 

Thanks very much for your valuable comments.

Reviewer 3 Report

Congratulations on your study. I had the following questions about the stimuli and the design of the experiment.

  1. For the contour variation tones, why use flat contours at three different F0s as standard? I can see how using the flat contour with an F0 of a 100 Hz for the set might provide a JND of pitch contour but I am not clear if its a JND of pitch contour when the subjects are comparing a 200 Hz flat F0 to a rising F0 starting at 100 Hz? The flat 200 Hz compared to a rising tone that started at 200 Hz would be a JND. Comparing a 200 Hz flat to a rising tone starting at 100 Hz would have many more cues and given it is speech it might have other cues that make it very different from listening to the flat starting at the same onset F0. Even your reference of the study with young adults (Huang et al., 2015) uses flat and contours as standards for JND measurements for flat and contour tones.
  2.  Why pick this particular age group of children? Are there other references to development of tonal processing in young normal children that you can use to support your choice of participants and use to explain your results?
  3. P8. L 262-265. I am not sure that the evidence supports your conclusion. I can understand the lack of maturity explanation, but I am not sure I follow this idea.
  4. Please use a table to summarize your ANOVA results. It will make the data much more accessible to a reader.

Round 2

Reviewer 1 Report

After corrections, the quality of the article has definitely improved. Now the manuscript meets all the standards of a scientific article and complies with editorial guidelines.

After corrections, the quality of the article has definitely improved. Now the manuscript meets all the standards of a scientific article and complies with editorial guidelines.
Minor corrections are required to complete the revision and make the article publishable.
1. The authors placed within the limits the impossibility of tracing the type of surgery. In any case, please write that technically the round window approach manages to preserve hearing residues and that theoretically the results could be better, but further studies are needed.
2. In the statistics section, which now appears clear, please specify the name of the tests performed for the comparison between groups (T studen? Mann whitney u test?).
What kind of correlation test did you use? Spearman or Person?)
